Marine or freshwater: the role of ornamental fish keeper’s preferences in the conservation of aquatic organisms in Brazil

Borges Anna Karolina Martins 1 karolm26@hotmail.com
Oliveira Tacyana Pereira Ribeiro 2
http://orcid.org/0000-0001-6824-0797 Alves Rômulo Romeu Nóbrega 3
1 Programa de Pós-Graduação em Etnobiologia e Conservação da Natureza, Universidade Federal Rural de Pernambuco , Recife , Brazil
2 Centro de Ciências Biológicas e Sociais Aplicadas, Universidade Estadual da Paraíba , João Pessoa , Brazil
3 Departamento de Biologia, Universidade Estadual da Paraíba , Campina Grande , Brazil
Reimer James
Electronic publication date: 2022 Nov 11
Publication date: 2022
Volume: 10
Electronic Location ID: e14387
Received 2022 Jul 18; Accepted 2022 Oct 21
Copyright: © 2022 Borges et al.
Copyright year: 2022
Copyright holder: Borges et al.
License: This is an open access article distributed under the terms of the Creative Commons Attribution License, which permits unrestricted use, distribution, reproduction and adaptation in any medium and for any purpose provided that it is properly attributed. For attribution, the original author(s), title, publication source (PeerJ) and either DOI or URL of the article must be cited.
License URL: https://creativecommons.org/licenses/by/4.0/

Keywords: Social media, Marine fish, Freshwater fish, Invasive species, Trade regulations

Funding: Brazilian National Council for Scientific and Technological Development 130751/2018-9 CNPq 422041/2018-1 FAPESQ-PB The Brazilian National Council for Scientific and Technological Development granted a MSc scholarship to Anna Karolina Martins Borges (CNPq, 130751/2018-9) and a productivity research grant to Rômulo Romeu Nóbrega Alves. This study was also supported by CNPq (grant 422041/2018-1) and FAPESQ-PB. The funders had no role in study design, data collection and analysis, decision to publish, or preparation of the manuscript.

==============================
Background

The use of ornamental fish as pets has important implications for the conservation of the species used in fish keeping, particularly in relation to overexploitation. Understanding ornamental fish keepers’ relationship with the hobby can provide important information for assessing the potential impacts of the activity. Here, we analyzed the profile of Brazilian ornamental fish keepers and evaluated their preferences and the implications of their choices.

Methods

Information was obtained by applying questionnaires to 906 ornamental fish keepers participating in fish keeping groups in a social network. The questionnaire contained questions about the species of fish kept (freshwater and marine), techniques used, socio-economic aspects, and associated conservation perspectives.

Results

Most ornamental fish keepers were young men (20–40 years old), with higher education and monthly income above US$ 530.00. Participants predominantly kept freshwater fish (86%), but marine fish only (5%) or both marine and freshwater hobbyists (9%) were also recorded. A total of 523 species of ornamental fish were kept, most of which comprised freshwater (76% of the total) and exotic species (73%). About a third of the fish species recorded were under national trade restrictions. In addition, about a third of ornamental fish keepers declared that they also had invertebrates. Marine aquariums require a greater financial investment, especially at the beginning, than freshwater aquariums and are also almost entirely based on exotic species. The aesthetic factor is the main motivation associated with practicing this hobby, being color and behavior key factors in choosing fish. A total of 10% of hobbyists have already released fish into the wild, highlighting concerns about potential biological invasions. There is an urgent need to enforce regulations towards restricting ornamental fish keepers’ access to threatened native species and potentially invasive species, as well as measures aimed at informing and raising hobbyists’ awareness of conservation measures related to the hobby.

Introduction

The ornamental use of animals, whether for entertainment or company, is currently considered one of the main hobbies in the world (Alves & Rocha, 2018; Alves et al., 2019), generating a global trade of about US$ 15–30 billion (Evers, Pinnegar & Taylor, 2019). The practice of raising fish for ornamental purposes, known as fish keeping, has been practiced for at least a thousand years (Stern et al., 2018; Walster et al., 2015; Novák, Kalous & Patoka, 2020; Novák et al., 2022) and has become a segment of the pet market which has continuously grown since 1970 at an annual rate of 14% (Evers, Pinnegar & Taylor, 2019).

Fish ranks third as the most popular pets in the world (Reynoso et al., 2012; Kumar, Gunasundari & Prakash, 2015; Pate et al., 2019). Fish keeping is also a very popular hobby in Brazil, with an estimated 19.1 million ornamental fish being kept as pets in 2018 (ABINPET, 2019). Considering only the domestic online trade, Brazilian fish keeping involves at least 609 species of marine and freshwater fish (Borges et al., 2021). In addition, Brazil occupies an important place in the world aquarium scene, being one of the main exporters of ornamental fish (Rhyne et al., 2012), reaching third place in the world market in the sector (ABINPET, 2019).

Consumer preferences are considered the major drivers of the ornamental market (Hinsley, Verissimo & Roberts, 2015; Magalhães et al., 2017; Sung & Fong, 2018). Ornamental fish keepers’ preferences, for instance, are linked to rare or threatened species, with striking colors, peculiar shapes, and/or behaviors (Dhar & Ghosh, 2015; Morcom et al., 2018; Borges et al., 2021). Furthermore, new preferences are emerging. For instance, sales of large-bodied ornamental species, known as “tankbusters”, have grown rapidly in the last decade (Holmberg et al., 2015; Magalhães et al., 2017). This applies to both marine and freshwater species, despite a much larger number of freshwater species being used for aquariums (Evers, Pinnegar & Taylor, 2019). It is estimated that aquarism worldwide involves more than 6,500 freshwater and 1,802 marine fish species (Raghavan et al., 2013; Novák, Kalous & Patoka, 2020), of which at least 80% are from the tropics (Stern et al., 2018). Besides fish, aquarism also involves a wide variety of invertebrate species, both marine and freshwater, encompassing more than 150 species of stony corals and hundreds of of non-coral invertebrate species, if we consider only the marine ornamental trade (Wabnitz et al., 2003; Rhyne et al., 2012).

Such motivations for ornamental fish keeping raise particular conservation concerns, as several of the species involved in the ornamental trade are vulnerable to overexploitation, such as the cardinal tetra Paracheirodon axelrodi (Schultz, 1956) and the zebra pleco Hypancistrus zebra Isbrücker & Nijssen 1991 in South America and Chromobotia macracanthus (Bleeker, 1852) in Asia (Sadovy & Vincent, 2002; Banha, Diniz & Anastácio, 2019; Evers, Pinnegar & Taylor, 2019). These concerns are also applied to exotic species (Patoka et al., 2016, 2018), since aquarism is considered an important source of invasive species, either through intentional or unintentional release (Magalhães et al., 2017; Maceda-Veiga et al., 2019; Patoka et al., 2020), the unintentional introduction of ‘hitchhiking’ species (Duggan, 2010; Patoka et al., 2016; Ložek, Patoka & Bláha, 2021), or the introduction of disease vectors (Maceda-Veiga et al., 2013; Magalhães & Jacobi, 2013; Putra et al., 2018). On the other hand, if fish keeping is well managed, it can be an ally for biodiversity conservation (Maceda-Veiga et al., 2016). In addition to being considered a crucial source of income for several local economies (Zehev et al., 2015; Evers, Pinnegar & Taylor, 2019), it can offer a possibility to protect endangered species by developing strategies for captive breeding and sustainable trade (Rhyne et al., 2017; Novák, Kalous & Patoka, 2020; Novák et al., 2022).

Understanding the trade-offs between hobbyists’ preferences and ornamental fish conservation may be an invaluable tool for advancing the protection and sustainable use of the exploited species (Hinsley, Verissimo & Roberts, 2015), especially in countries that play an essential role in the trade, such as Brazil. Here, we evaluated the preferences for marine and freshwater ornamental fish species by Brazilian aquarium hobbyists from Facebook groups. We assessed Facebook as it comprises one of the most popular online platforms worldwide where users feel comfortable sharing their ideas and experiences about the hobby (Pi, Chou & Liao, 2013) and has been identified as one of the most used in the wild animal trade (Magalhães et al., 2017; Haysom, 2018; Sy, 2018). We aimed to: (1) identify the motivations behind hobbyists’ choices in keeping fish as pets; (2) describe the hobbyists’ profile and their behavior in relation to the hobby and animals; (3) relate their preferences and behaviors to conservation issues; and (4) compare all these aspects between marine and freshwater ornamental fish keepers.

Materials and Methods

Data collection

Data were gathered through online questionnaires applied to 906 aquarium hobbyists from six Brazilian aquarism groups on the Facebook social network (https://www.facebook.com/), between January and March 2019. For determining the Facebook group to be sampled, we performed searches using the keywords fishkeeping, fish+ornamental, aquarium+ornamental (in Portuguese, aquarismo, peixes+ornamentais, aquário+ornamentais), considering that they bring together people who keep ornamental fish and use the social network to exchange information about the hobby. Our search resulted in 83 groups, of which six were selected for being more active (higher frequency of publications). Then, we individually contacted all profiles (n = 4,630) from those six groups over 3 months through private messages (Alves et al., 2019), in which the objectives of the study were explained, and the ornamental fish keeper was invited to answer a semi-structured questionnaire (Albuquerque et al., 2014). Once the invitation was accepted and the person expressed in writing their consent to participate in the research, each participant received a new message containing the link that directed them to the online questionnaire. Despite sent by private message to those interested in take part of the research, questionnaires remained anonymous, as no personal information (i.e., name, email contact) was collected from the participants. The questionnaire contained closed, open, and mixed questions about socio-economic aspects and the hobby (see Supplemental Material). All questionnaires were applied online using the Google Forms platform tool (https://www.google.com/intl/pt-BR/forms/about/). The study was approved by the Research Ethics Committee of Hospital Universitário Lauro Wanderley (CEP/HULW, authorization # 3.062.563).

The income of the research participants was determined based on the Brazilian minimum monthly wage in force at the time of data collection. These values were converted to US dollars considering the exchange rate for the period in which the data were collected (Brazilian minimum wage in 2019: US$ 1 to BR$ 3,71).

Species were identified based on participants’ information and confirming the photographs sent by them. We used the Fishbase database (Froese & Pauly, 2019) and consulted specialists for those species whose identification was problematic. To verify if the recorded species were allowed to be traded for ornamental purposes and their conservation status, we consulted the legislation applied in Brazil during the period covered by this study, as previously described in Borges et al. (2021), and also the Red List of the International Union for the Conservation of Nature (IUCN, 2019), and the list of the Convention on International Trade in Endangered Species of Wild Flora and Fauna (CITES, 2019).

Data analysis

We used descriptive statistics to describe the profile of survey participants and their responses to aquarium-related questions. Percentages were calculated based on the number of citations. Thus, the sum of n may exceed the total number of participants (n = 906) in case of the questions allowed more than one answer option. Fisher’s exact test was used to assess associations between socioeconomic factors (e.g., gender, age, monthly income) and hobby characteristics (e.g., marine-only or freshwater-only ornamental fish keepers, number of fish kept, experience time in the aquarium hobby, monthly spending on the hobby), with a significance threshold of p < 0.05. For analyzes comparing responses from freshwater-only and marine-only ornamental fish keepers, responses from hobbyists who claimed to have both types of aquariums were not considered. We also used a word cloud analysis to visually evaluate ornamental fish keepers’ perceptions regarding the release of ornamental fish in the wild. This technique makes it possible to identify which words were most frequently cited by the participants (McNaught & Lam, 2010; Barbosa-Filho et al., 2020). All analyzes were performed using the R software version 3.6.1 (R Core Team, 2019).

Results

Profile of Brazilian ornamental fish keepers

Most ornamental fish keepers were male (88%; n = 795), aged between 20 and 40 years (73%; n = 661). Approximately two-thirds of ornamental fish keepers (69%; n = 627) had higher education and the monthly income of the majority (78%; n = 744) was over US$ 530 (equivalent to two times the Brazilian minimum wage). Ornamental fish keepers residing in 24 Brazilian states participated in the study, most of them from the southeastern region (51%; n = 468), mainly from the states of São Paulo (40%; n = 308) and Rio de Janeiro (12.8%; n = 116; Fig. 1). Most ornamental fish keepers had practiced the hobby for less than 5 years (40%; n = 372) and had mainly acquired experience through internet searches (corresponding to 22% of the sources mentioned; Fig. 2).

Figure 1 Socioeconomic characteristics of research participants (n = 906 aquarists) and their distribution in Brazilian states.

Abbreviation of Brazilian states: AC, Acre; AL, Alagoas; AM, Amazonas; AP, Amapá; BA, Bahia; CE, Ceará; ES, Espírito Santo; GO, Goiás; MA, Maranhão; MG, Minas Gerais; MS, Mato Grosso do Sul; MT, Mato Grosso; PA, Pará; PB, Paraíba; PE, Pernambuco; PI, Piauí; PR, Paraná; RJ, Rio de Janeiro; RN, Rio Grande do Norte; RO, Rondônia; RR, Roraima; RS, Rio Grande do Sul; SC, Santa Catarina; SE, Sergipe; SP, São Paulo; TO, Tocantins.

Figure 2 Answers from ornamental fish keepers interviewed to questions about their relationship with the hobby.

Most hobbyists prefer to acquire their animals through physical stores or pet shops (55% of the sources mentioned) and own freshwater aquariums only (86%; n = 776). We also recorded participants who kept both marine and freshwater aquariums (9%; n = 84).

Animals used in the hobby

Participants kept fish belonging to 523 species (93 families), most of them freshwater (76%; n = 400 species). The beauty of aquariums and fish (76%; n = 790) was the main motivation for practicing aquarism (Fig. 2). Behavior (26%; n = 537) and aesthetics (22%; n = 441) were key factors in choosing fish for the hobby (Fig. 2). Most species were not native to Brazil (73%; n = 293)—half of the freshwater species (50%; n = 203) and more than half of marine species (73%; n = 90). In addition, 33% (n = 131) of the species had their exploitation prohibited in the country (see Supplemental Material).

Ornamental fish keepers preferred the following freshwater fish families: Cichlidae (34%), Characidae (12%), and Cyprinidae (11%), especially the following species: Pterophyllum scalare (4.1%), Hypostomus plecostomus (3.7%) and Poecilia reticulata (3.5%), all native to Brazil. The preferred families regarding marine fish were Acanthuridae (28%) and Pomacentridae (28%), with a predilection for the following species: Amphiprion ocellaris (18%), Zebrasoma flavescens (9%), and Paracanthurus hepatus (7%), all exotics. Thirty-nine species were included in threatened categories in the IUCN classification, namely: 10 species are listed as “Near Threatened”, 14 species as “Vulnerable”, eight species as “Endangered”, and seven species as “Critically Endangered” (Table 1). In addition, six species were included in the CITES Appendices and five species are included in the Brazilian List of Threatened Species (MMA, 2014) (see Supplemental Material).

Table 1 Ornamental fish species kept by Brazilian aquarists with some degree of threat according to the IUCN classification.

Species	Family	25+ cited	Native	Brazilian legislation	
Neart threatened					
Altolamprologus calvus (Poll, 1978)	Cichlidae		No	Allowed	
Aulonocara nyassae Regan, 1922	Cichlidae		No	Allowed	
Corydoras panda Nijssen & Isbrücker, 1971	Callichthyidae		Yes	Allowed	
Chiloscyllium punctatum Müller & Henle, 1838	Hemiscylliidae		No	Allowed	
Chindongo elongatus (Fryer, 1956)	Cichlidae		No	Allowed	
Chitala chitala (Hamilton, 1822)	Notopteridae		No	Forbidden	
Chitala blanci (d’Aubenton, 1965)	Notopteridae		No	Allowed	
Danio kyathit Fang, 1998	Cyprinidae		No	Allowed	
Erpetoichthys calabaricus Smith, 1865	Polypteridae		No	Forbidden	
Trichopodus leerii (Bleeker, 1852)	Osphronemidae		No	Allowed	
Vulnerable					
Aulonocara hansbaenschi Meyer, Riehl & Zetzsche, 1987	Cichlidae		No	Allowed	
Aulonocara steveni Meyer, Riehl & Zetzsche, 1987	Cichlidae		No	Forbidden	
Balantiocheilos melanopterus (Bleeker, 1850)	Cyprinidae		No	Allowed	
Cyprinus carpio Linnaeus, 1758	Cyprinidae	Yes	No	Allowed	
Cyrtocara moorii Boulenger, 1902	Cichlidae		No	Allowed	
Datnioides undecimradiatus (Roberts & Kottelat, 1994)	Datnioididae		No	Forbidden	
Haplochromis aeneocolor Greenwood, 1973	Cichlidae		No	Forbidden	
Haplochromis obliquidens (Hilgendorf, 1888)	Cichlidae		No	Allowed	
Notholebias minimus (Myers, 1942)	Rivulidae		Yes	Forbidden	
Paratilapia polleni Bleeker, 1868	Cichlidae		No	Forbidden	
Pethia nigrofasciata (Günther, 1868)	Cyprinidae		No	Forbidden	
Pseudotropheus demasoni (Konings, 1994)	Cichlidae		No	Allowed	
Puntius titteya Deraniyagala, 1929	Cyprinidae		No	Forbidden	
Tropheus duboisi Marlier, 1959	Cichlidae		No	Allowed	
Endangered					
Champsochromis spilorhynchus (Regan, 1922)	Cichlidae		No	Forbidden	
Glossolepis incisus Weber, 1907	Melanotaeniidae		No	Allowed	
Melanotaenia boesemani Allen & Cross, 1980	Melanotaeniidae		No	Allowed	
Pangasianodon hypophthalmus (Sauvage, 1878)	Pangasiidae	Yes	No	Forbidden	
Placidochromis phenochilus (Trewavas, 1935)	Cichlidae		No	Allowed	
Pterapogon kauderni Koumans, 1933	Apogonidae	Yes	No	Allowed	
Puntius arulius (Jerdon, 1849)	Cyprinidae		No	Allowed	
Sahyadria denisonii (Day, 1865)	Cyprinidae		No	Allowed	
Critically endangered					
Aulonocara baenschi Meyer & Riehl, 1985	Cichlidae		No	Allowed	
Aulonocara maylandi Trewavas, 1984	Cichlidae		No	Allowed	
Chindongo saulosi (Konings, 1990)	Cichlidae		No	Forbidden	
Epalzeorhynchos bicolor (Smith, 1931)	Cyprinidae		No	Allowed	
Haplochromis latifasciatus Regan, 1929	Cichlidae		No	Forbidden	
Melanochromis chipokae Johnson, 1975	Cichlidae		No	Allowed	
Pseudotropheus cyaneorhabdos (Bowers & Stauffer, 1997)	Cichlidae		No	Forbidden	

Besides keeping fishes, about a third of ornamental fish keepers (36%; n = 328) declared having invertebrates, mentioning 66 animals belonging to 14 classes, mainly crustaceans (48%; n = 28) and mollusks (33%; n = 22) (see Supplemental Material). Invertebrate maintenance was more common among marine (70% of marine ornamental fish keepers) than among freshwater ornamental fish keepers (30%). The most common invertebrates kept by the participants were freshwater snails of the genus Pomacea (23%) and shrimps of the genus Neocaridina (16%). Of the 33 species identified, 22 (66%) were non-native to Brazil, including species with introduction records (n = 7), in Brazil or in other countries (e.g., Procambarus clarkii (Girard, 1852), Melanoides tuberculata (O.F. Müller, 1774), Artemia salina (Linnaeus, 1758), Physella acuta (Draparnaud, 1805)).

Maintenance of the animals

Most participants claimed to keep between 10 and 50 individuals (64%; n = 675; Fig. 3), mostly in mixed-species tanks (34%; n = 512; Supplemental Material). The number of fish kept by hobbyists was related to income (p < 0.001) and with the experience time in the aquarium hobby (p = 0.003). Most participants (52%; n = 475) claimed to spend less than US$ 20 monthly to maintain the hobby (Fig. 3). Monthly costs were not related to the number of fish kept by the participants (p = 0.07), but to the habitat type of the animals (marine or freshwater) (p < 0.001), and ornamental fish keepers who own marine fish reported higher expenses (Fig. 3).

Figure 3 Answers from Brazilian ornamental fish keepers questioned about aspects related to hobby maintenance.

Most participants (40%; n = 364) stated that they cleaned aquaria and performed partial water changes weekly (36%; n = 330) or biweekly (30%; n = 273). Fish illness was reported by 74% of the participants (n = 675), who mentioned 26 types of diseases, the most cited being: ictio (white spots disease caused by protozoan Ichthyophthirius multifiliis; n = 374), diseases caused by fungi (n = 72), and hole in the head (caused by the protozoan Hexamita intestinalis; n = 35). The treatments used were mainly based on exchanging experiences with other ornamental fish keepers (40%; n = 419; Fig. 3) and involved the use of antibiotics, fungicides, and changes in the physico-chemical parameters of the aquarium (e.g., temperature, salinity, luminosity; see Supplemental Material).

When asked about their attitude when they want to dispose of a fish, the most frequent attitude was donation (44%; n = 559). Other attitudes mentioned were discussions with other ornamental fish keepers (30%; n = 382), sale (22%; n = 275), released in the wild—aquarium dumping (2%; n = 30), exchange or return at the store where they purchased the animal (0.2%; n = 3) and use for feeding (0.1%; n = 2).

Collecting and releasing fish in the wild

About 10% (n = 95) of participants collected fish in the wild, mostly freshwater-only ornamental fish keepers (77%; n = 73). Among those, most had no formal education (p < 0.001; 38%), kept more than 500 fish (p < 0.001; 42%), and declared that they had no expenses to maintain the hobby (p = 0.007; 34%; Table 2). The collections were always carried out in habitats close to the hobbyists’ homes (e.g., rivers, streams, waterfalls, creeks, lagoons/lakes, ponds, dams, estuaries, beaches, rocky shores), using different techniques (i.e., hook and bait, fishing net, sieve, cast net, artisanal traps, manual collection). Fish collections belonging to at least 19 families were reported, mainly Cichlidae (32%) and Poeciliidae (18%).

Table 2 Association between socioeconomic characteristics from Brazilian ornamental fish keepers and the capture and release of ornamental fish in the wild (significance: *p < 0.05, **p < 0.01, ***p < 0.001).

Independent variables	Capture in the wild	Release in the wild	
Yes (%)	No (%)	p-value	Yes (%)	No (%)	p-value	
Gender			0.01*			0.0009***	
Female	4	96		0	96		
Male	11	89		11	85		
Not declared	50	50		0	100		
Age group			0.19			0.02*	
Less than 20 years	18	82		21	75		
Between 20 and 30 years	11	89		11	84		
Between 30 and 40 years	10	90		8	89		
Between 40 and 50 years	8	92		4	90		
More than 50 years	11	89		6	84		
Monthly income			0.02*			0.01*	
Less than US$ 266	18	82		20	80		
Between US$ 266–US$ 530	16	84		12	81		
Between US$ 530–US$ 800	10	90		8	87		
Between US$ 800–US$ 1,060	8	92		8	88		
Between US$ 1,060–US$ 2,660	8	92		8	90		
More than US$ 2,660	25	75		25	75		
Not declared	9	91		13	76		
Education level			0.0009***			0.11	
No formal education	38	62		25	75		
Elementary school	21	79		20	75		
High school	9	91		9	86		
Incomplete higher education	11	89		8	89		
Complete higher education	6	94		9	85		
Postgraduate studies	10	90		8	90		
Hobby experience			0.03*			0.03*	
Less than 1 year	7	93		6	90		
Between 1 and 5 years	8	92		7	90		
Between 5 and 10 years	9	91		10	86		
Over tem years	8	92		9	86		
Since childhood	15	85		15	80		
Number of fish			0.0009***			0.28	
Less than 10 fish	7	93		11	84		
10–20 fish	8	92		8	88		
20–50 fish	13	87		9	88		
50–100 fish	20	80		7	86		
100–500 fish	8	92		8	84		
More than 500 fish	42	58		26	69		
Monthly spend			0.007**			0.42	
Don’t spend nothing	34	66		33	67		
Less than US$ 20	9	91		9	87		
Between US$ 200–US$ 100	10	90		10	85		
Between US$ 100–US$ 200	26	74		14	84		
Between US$ 200–US$ 400	14	86		14	86		
More than US$ 400	0	100		12	88		

Considering releases, 10% (n = 88) of participants revealed they had released animals from their aquaria into the wild, most of them freshwater-only ornamental fish keepers (90%; n = 80). Regarding the profile of participants who release fish in the wild, those most likely to perform this practice were males (p < 0.001; 11%), with less than 20 years (p = 0.02; 21%), and with an income of more than 10 Brazilian minimum wages (p = 0.01; 25%; Table 2). The releases were made in rivers, streams, waterfalls, creeks, springs, and lakes close to their homes; in artificial lakes and ponds in public parks; in private dams and weirs; and into the sea. Releases of fish belonging to at least nine families have been reported, mainly Cichlidae (29%) and Loricariidae (22%), and invertebrates of two classes, Gastropoda (Pomacea spp.) and Malacostraca (Macrobrachium carcinus). Participants who said they had never released any animals (86%; n = 780) expressed their opinions regarding this practice and the most cited words were: “never”, “release”, “nature”, “species”, “fish” and “native” (Fig. 4). Asked about the future of the ornamental fish trade, 66% (n = 599) stated that more fish will be available for commercialization due to cultivation and 34% (n = 307) believe that less fish will be available due to overfishing.

Figure 4 Wordcloud elaborated from ornamental fish keepers’ responses who are against the practice of releasing ornamental fish in the wild (n = 780 responses).

Larger words indicate higher frequencies of occurrence.

Discussion

Hobby overview

The profile of the Brazilian hobbyist is quite homogeneous, being mainly formed by men with education and monthly income above the general average in Brazil (most of the adult population has at most completed elementary school and a monthly income equivalent to little more than two minimum wages; IBGE, 2019). This profile with an above-average financial situation reinforces that fish keeping is a luxury hobby (Rhyne & Tlusty, 2012), although technological development has facilitated maintaining ornamental fish and increased the popularity of the activity (Arbuatti & Lucidi, 2010; Banha, Diniz & Anastácio, 2019). Despite this, fish keeping is a hobby that could demand a great financial commitment, since the acquisition of fish, equipment (e.g., hang-on/canister filters), and materials for aquaria maintenance (e.g., food and water conditioners) can demand a high investment, which often goes beyond what the majority of the population can afford.

As expected, aesthetics was the main motivation for choosing fish keeping as a hobby. Fish with striking color patterns and different shapes attract people and make them want to have those fish at home for contemplation and entertainment (Walster et al., 2015). Hobbyists in our study also highlighted as a motivation keeping fish for therapeutic purposes, a trend registered in studies associated with pets of different taxa, which revealed the positive impact pets can have on the quality of life of people who have stress or depression (Enmarker et al., 2012; Brooks et al., 2016, 2018; Hui Gan et al., 2020). The benefits from the contact with elements of natural environments for health and well-being (i.e., biophilia) has already been demonstrated (Grinde & Patil, 2009; Engemann et al., 2021), and regarding fish keeping, this is a factor that has contributed to popularizing the hobby, attracting people who claim seek relaxation and reduced stress levels (Magalhães et al., 2017; Stern et al., 2018).

Freshwater fish dominate fish keeping in Brazil, representing more than half of the species kept by hobbyists who participated in this study. This preference for continental fish was also recorded by Mazza et al. (2015) in a study performed in Italy, and can mainly be attributed to price, as freshwater fish tend to be cheaper than marine fish. Another factor pointed out by these authors which may be decisive for this preference is the ease of maintenance, considering that maintaining aquaria with marine fish requires greater dedication and improved equipment, in addition to higher financial investment, as we were also able to verify with our results (Borges et al., 2021).

In addition to fish, invertebrates are also exploited by ornamental fish keepers in Brazil, although to a lesser extent. Despite the use of invertebrates not being as well documented as the use of fish in fish keeping, it has been recorded by some studies which have highlighted the popularized trade of these animals for this purpose (Ng et al., 2016; Uderbayev et al., 2017; Alves & Rocha, 2018; Akmal et al., 2020). Without considering corals, it is estimated that about 500 species of invertebrates are used in fish keeping, including mollusks (gastropods, bivalves, and cephalopods), polychaetes, crustaceans (shrimps, crabs, and lobsters), anemones and echinoderms (starfish, urchins) and sea cucumbers (Palmtag, 2017; Alves & Rocha, 2018). In addition to the aesthetic factor, invertebrates are sought after by ornamental fish keepers interested in the useful services they can provide. Integrating so-called “cleaning teams”, the invertebrates work by eliminating parasites from the aquarium, filtering the water, and sifting the substrate (Palmtag, 2017; Patoka et al., 2017).

Several factors are crucial to ensure the well-being of fish kept as ornamentals, and almost all of them directly depend on the hobbyist’s knowledge (Walster et al., 2015). Herein, we found that ornamental fish keepers know the needs of the organisms they maintain and adapt aspects such as cleaning and partial water changes to these needs, taking into account the different needs of their animals (marine fish, freshwater fish, and invertebrates). Some of the main difficulties in keeping ornamental fish are related to water quality maintenance, including parameters such as temperature, pH, GH (general hardness), KH (carbonate hardness), salinity, ammonia, nitrate/nitrite concentration, and dissolved oxygen (Stevens et al., 2017). In addition, inadequate diets, handling, crowding, living with fish species with different needs, the aggressiveness of some species, and the confinement itself are factors that can cause stress to fish, often culminating in death (Walster et al., 2015; Stevens et al., 2017).

An imbalance in any of the necessary factors for the well-being of fish in captivity can negatively affect resistance to pathogens, leading to the development of diseases (Cardoso et al., 2019). Most hobbyists reported the occurrence of diseases in their fish for which they mentioned a series of treatments, mostly carried out on their own based on information obtained from other ornamental fish keepers or on the internet. The lack of access to a diagnosis performed by professionals, whether due to logistical or financial difficulties, besides the fact that medicines for ornamental aquatic organisms can also be expensive, can lead many hobbyists to use over-the-counter remedies and home treatments, which can have the opposite effect, causing more suffering to the animals (Walster et al., 2015). This issue is even more challenging for aquarium invertebrates, as treatment is impossible or unknown in many cases (Dombrowski & De Voe, 2007). Home treatments for diseases in ornamental fish still raise a concern related to public health. The substances used for these treatments, such as antibiotics, fungicides, and parasiticides, are often added to the aquarium water, and if this untreated water is discarded into the sewage system it can contaminate water that is used by the human population.

Implications of the hobby for conservation

Collecting fish in the wild to meet the demands and preferences of ornamental fish keepers raise concerns about the long-term sustainability of this practice (Evers, Pinnegar & Taylor, 2019). In addition to the overexploitation of species, destructive fishing practices and the high risk of biological invasion can transform ornamental fish keeping into a threat to biodiversity (Magalhães et al., 2017; Patoka et al., 2018; Banha, Diniz & Anastácio, 2019; Evers, Pinnegar & Taylor, 2019). Most ornamental fish keepers in our study believed that a greater number of ornamental fish will be available in the market in the future due to improved cultivation and breeding practices in captivity. In fact, aquaculture has been touted as the solution to the over-exploitation of species for ornamental purposes (Teletchea, 2016). However, it cannot be treated as a panacea, as most of the successful cultivation has only been developed for freshwater fish, while the marine ornamental fish trade almost entirely depends on collection in the wild (Máñez, Dandava & Ekau, 2014; Morcom et al., 2018).

Ornamental fish are among the most introduced aquatic animals worldwide (Magalhães & Jacobi, 2013) and the ease of acquiring exotic species through pet shops and the internet enhances the ability of ornamental fish keepers to trigger introductions (Azevedo-Santos et al., 2015; Magalhães et al., 2017). The action of invasive species is considered the second leading cause of extinction worldwide (Wilcove et al., 1998). These species act by drastically modifying the ecosystems in which they settle, causing the decline of native species and resulting in biodiversity loss (Mantelatto et al., 2018). All these alterations can influence beta diversity patterns (i.e., β-diversity), causing biotic homogenization (Magalhães et al., 2021). There are introduction records of more than 150 non-native fish species in the last two centuries in Brazil, many of which were introduced through fish keeping (Azevedo-Santos et al., 2015; França et al., 2017; Magalhães et al., 2017; Magalhães et al., 2021).

In addition to unintentional escapes, mainly caused by inappropriate handling and keeping (Calado & Chapman, 2006), hobbyists can also promote the intentional release of ornamental aquatic animals. Here, we found that 10% of fish keepers have already discarded ornamental fish in natural environments. A similar result was found by another study in Brazil and this finding is especially alarming because, if projected for all ornamental fish keepers existing in the country, it might represent a high propagule pressure (Geller et al., 2020). Hobbyists discarding their fish in the wild is mainly motivated by the excessive growth of the animals, a demonstration of aggression, or the time and money demand that the hobby requires (Magalhães & Jacobi, 2013; Azevedo-Santos et al., 2015; Magalhães et al., 2017). For example, some of the fish genera kept by the ornamental fish keepers interviewed, such as Channa Scopoli 1777, Pangasianodon Chevey 1931, Clarias Scopoli 1777 and Osphronemus Lacepède 1801, can reach large sizes and exhibit predatory behavior, representing a high risk of biological invasion (Magalhães & Jacobi, 2013; Magalhães et al., 2017) and are commercialized for ornamental purposes with restrictions in Brazil (MPA, 2012). Other examples of aquarium introductions (i.e., aquarium dumping) include the jaguar cichlid (Parachromis managuensis) in northeastern Brazil (França et al., 2017) and the lionfish (Pterois volitans) along the east coast of North America (Whitfield et al., 2002) and in Brazil (Ferreira et al., 2015), with the latter being considered one of the worst marine invasive species.

Invertebrates are also of concern when they come to biological invasion. The invertebrates most used by ornamental fish keepers participating in the study were snails of the genus Pomacea, which had their importation prohibited in the European Union because they represent a high threat of invasion (Ng et al., 2016). Herein, we recorded the release of these animals into the wild. We also recorded the use of some other species that were previously introduced in Brazil and other countries through fish keeping. For example, the Thiarid snail (Melanoides tuberculatus), which was first recorded in Brazil in 1967 and currently has extensive distribution throughout the country (Barros et al., 2019). It was primarily introduced and dispersed through the plant and animal trade (Vaz et al., 1986), with the high abundance of this snail having caused damage to the native benthic community of Brazil (Pointier, 1993). Another example is the case of the Red swamp crayfish (Procambarus clarkii) which has established itself worldwide after being extensively translocated by the aquarium trade (Torres & Álvarez, 2012) and is considered one of the 100 worst invaders in Europe (Gherardi, 2007). The establishment of this species has already been demonstrated in Brazil (Loureiro et al., 2015) with evidence of negative impacts on native fauna (Banci et al., 2013) and it has been hypothesized that the introduction occurred through fish keeping (Magalhães et al., 2005; Magalhães & Andrade, 2014). Despite imports, transportation, and commercialization of P. clarkii being prohibited in the country for more than two decades (IBAMA, 2008), a recent study showed that the species is still freely commercialized by aquarium stores (Smialoski & Almerão, 2022), which makes clear the ineffectiveness of this legislation.

Besides its potential to promote intentional introductions, fish keeping also acts as a dispersion route or “stepping stones” for other potentially invasive species that can be accidentally transported along with fish, such as invertebrates, aquatic plants, and microorganisms (Assis, Cavalcante & Brito, 2014; Patoka et al., 2016; Banha, Diniz & Anastácio, 2019). Zoonotic pathogens that pose risks to the pet industry and humans can also be spread through improper disposal of water used in aquariums (Cardoso et al., 2019; Pate et al., 2019).

Brazil is a signatory country of the United Nations Convention on Biological Diversity, a United Nations treaty, through which it commits to making efforts to prevent and control possible introductions in aiming to protect against the threat that invasions can pose to biodiversity (França et al., 2017). The regulation of fish importation for ornamental purposes in the country is based on lists of permitted species, with emphasis on species that are strictly prohibited due to high invasion risk. However, lack of disclosure of legal restrictions, existing—but unenforced—laws (i.e., ‘dead letters’), and precarious inspection prevent the full functioning of the legislation, in addition to the fact that there are ornamental fish keepers willing to violate the law to maintain freshwater and marine prohibited species (Marchio, 2018; Patoka et al., 2018; Banha, Diniz & Anastácio, 2019).

Management and conservation opportunities

Most ornamental fish keepers participating in the study are aware of the risks related to the release of non-native fish into the wild, reinforcing the idea that education and information campaigns are the main key to modifying behaviors that can trigger biological invasions. As fish keeping offers an opportunity to bring the population closer to scientific concepts by directly observing organisms and their ecological processes, the educational potential of the hobby is an opportunity that can be used to communicate conservation, seeking to sensitize hobbyists on the problem of biological invasions and involve them in conservation initiatives (Azevedo-Santos et al., 2015; Maceda-Veiga et al., 2016; Marchio, 2018; Evers, Pinnegar & Taylor, 2019).

Some strategies for disseminating this information are to distribute informative material in aquarium stores, discuss this subject in the specialized media (i.e., aquarium magazines and websites), and disseminate it on social networks (e.g., Instagram, Twitter, Facebook, WhatsApp groups) through channels of the environmental agencies themselves (Azevedo-Santos et al., 2015; Magalhães, Azevedo-Santos & Pelicice, 2021). The attitude of sellers is also a key point to inhibit improper disposal of ornamental fish through responsible sales. Consumers should receive all technical information about the species they intend to acquire, especially regarding behavior and maximum size (Magalhães & Jacobi, 2013; Azevedo-Santos et al., 2015). In addition to investing in environmental education, creating collection centers to receive unwanted animals offers an alternative to discarding fish in nature (Banha, Diniz & Anastácio, 2019); however, this alternative can be problematic in terms of receiving capacity at these locations and therefore must be strictly well planned. Another measure pointed out by experts is the euthanasia of unwanted fish, as a more rigorous and effective strategy to control intermittent “propagule pressure” and thus interrupt the cycle of intentional introductions (Magalhães et al., 2017; Patoka et al., 2018). Obviously, an effort is needed to review and readjust existing laws that are not enforced, publicize and make these laws accessible, and ensure proper inspection regarding compliance with these legal restrictions (Patoka et al., 2018).

Conclusions

Our results reveal the main trends in fish keeping in Brazil, such as the choice of hobby mainly being motivated by aesthetic and behavioral factors of the animals, the preference for freshwater species and non-native species, and the fact that most ornamental fish keepers are males with above-average education and financial situation. Furthermore, our findings support the premise that fish keeping has the potential to trigger biological invasions, especially considering that most species used in the hobby are exotic and many are known to be potential invaders. These results reinforce that performing fish keeping as a sustainable hobby, which does not represent a threat to biodiversity, requires the joint action of ornamental fish keepers, traders (i.e., importers, wholesalers, retailers, aquarium store owners), and environmental agencies. Considering that there are already legislations in the country that prohibits the use of certain species for fish keeping, efforts must now be directed toward environmental education/awareness practices and improved intermittent inspections and monitoring.

Supplemental Information

Supplemental Information 1 Raw data.

Click here for additional data file.

Supplemental Information 2 Supplementary material.

Click here for additional data file.

Supplemental Information 3 Questionnaire in the original language.

Click here for additional data file.

We thank all the participants. We thank Walter Lechner, Henrique A. C. Ramos and Luiz A. Rocha for their help in confirming the identification of fish species. We also thank the editors, James Reimer and Diogo B. Provete, and the reviewers, Jiří Patoka and the anonymous reviewer, for their valuable contributions.

Additional Information and Declarations

Competing Interests

Author Contributions

Human Ethics

Data Availability

The authors declare that they have no competing interests.

Anna Karolina Martins Borges conceived and designed the experiments, performed the experiments, analyzed the data, prepared figures and/or tables, authored or reviewed drafts of the article, and approved the final draft.

Tacyana Pereira Ribeiro Oliveira conceived and designed the experiments, authored or reviewed drafts of the article, and approved the final draft.

Rômulo Romeu Nóbrega Alves conceived and designed the experiments, authored or reviewed drafts of the article, and approved the final draft.

The following information was supplied relating to ethical approvals (i.e., approving body and any reference numbers):

The study was approved by the Research Ethics Committee of Hospital Universitário Lauro Wanderley (CEP/HULW, authorization # 3.062.563).

The following information was supplied regarding data availability:

All raw data are available in the Supplemental Files.

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
