# Peer review of "Marine or freshwater: the role of ornamental fish keeper’s preferences in the conservation of aquatic organisms in Brazil"

_PeerJ, doi:10.7717/peerj.14387_

## Round 0.1 · original submission · Major Revisions

Note from new Associate Editor (James Reimer) - I have taken over as Associate Editor (AE) on your paper from the previous AE, and will handle your paper from here on. I have gone over your paper, the submitted reviews, and can agree with the decision text by the former AE, as included below. Please note I have made a decision of "Major revisions" as I may send your paper out for one more round of review after submission, but I also agree that almost all of the reviewer and former AE requests seem fair and constructive. I hope you can respond in a detailed manner, and look forward to seeing your revised work.


Message from former AE:

I have now received two very positive reviews on your manuscript. Notice both reviewers also provided further comments directly in the pdf, so please refer to them when preparing your revision. Most of them were minor and authors shouldn't have much trouble addressing them.

I also saw the paper with great enthusiasm and I expect it to be highly cited, given the depth and breath of its scope. Authors successfully compiled and analysed data from a questionnaire, preparing good-looking figures.

My only concern is that a Chi-square test is actually not the best one in your case. It's only used to test for proportional data when you have a 2x2 matrix (for larger matrices you usually use a G-test) and at least in which all the expected frequencies should be at least 5%. This seems not hold in some of the tests. But in your case, when you clearly have predictor variables (e.g., age) and responses that are counts (e.g., number of fish kept), why not simply use a GLM for each response variable, with the proper distribution? You could even reduce the number of tests because you can include all predictor variables at once in the model.

·

Basic reporting

Dear Authors,

I read the paper titled "Marine or freshwater: hobbyists’ preferences and its role in ornamental fish conservation in Brazil" with interest. I found the text to be an interesting and nice addition to our knowledge about ornamental aquaculture and related risks. There are my comments and suggestions highlighted directly in the PDF copy. These are not plenty and easy to follow. Check the format of the references carefully. I found the information about invertebrates like P. clarkii to be very important (the modification of the text in this regard is suggested). I suggested also some new citations (partly of my authorship - simply because ornamental aquaculture is my main focus and we published a series of focused papers - but I am sure that you can find other suitable papers if you wish). Generally, I see the merit of the study and recommend the acceptance for publication after minor revision.

Sincerely,
Jiří Patoka

Experimental design

The research question was well-defined and clear. The design of the survey is suitable and well-described. The only query is: was the questionnaire anonymous?

Validity of the findings

The described findings align with previously published similarly focused studies and underline the importance of further education of the general public about risks related to ornamental aquaculture. The results support conclusions.

Additional comments

For your consideration: If the discussion about the invertebrates will be more comprehensive, what about modifying the title in this regard? The paper would be of a wider audience, I guess.

Reviewer 2 ·

Basic reporting

Clear and unambiguous, professional English used throughout.

Literature references, sufficient field background/context provided (I SUGGESTED MORE REFERENCES TO ENRICH THE MANUSCRIPT).

Professional article structure, figures, tables. Raw data shared.

Self-contained with relevant results to hypotheses

Experimental design

Original primary research within Aims and Scope of the journal.

Research question well defined, relevant & meaningful. It is stated how research fills an identified knowledge gap

Rigorous investigation performed to a high technical & ethical standard.

Methods described with sufficient detail & information to replicate.

Validity of the findings

Meaningful replication encouraged where rationale & benefit to literature is clearly stated.

All underlying data have been provided; they are robust, statistically sound, & controlled.

Conclusions are well stated, linked to original research question & limited to supporting results.

Additional comments

It is a very important manuscript where the authors analyzed the profile of Brazilian aquarists and evaluated their preferences and their consequences from the conservationist point of view.
One of the most important results of this excellent study was indicate that most aquarists were young men, with higher education and monthly income, clearly showing that aquarism is still an accessible hobby for few people in Brazil. Participants predominantly kept freshwater fish. A total of more than 500 species of ornamental fish were kept by hobbyists, most of which comprised freshwater and exotic species. About a third of the species recorded were under national trade restrictions. Marine aquariums require a greater financial investment than freshwater aquariums and are also almost entirely based on exotic species. The aesthetic factor is the main motivation associated with practicing this hobby, being color and behavior key factors in choosing fish. Aquarium hobbyists have already released fish into the wild, highlighting concerns about potential biological invasions.
These are very interesting and important results for the understanding of the hobby in Brazil. The authors did a good job performing a kind of X-RAY on Brazilian aquarism/aquarium trade! So, I recognize the effort and dedication and I applaud the initiative to obtain information to prepare this manuscript submitted to PeerJ! The manuscript is very well written and the figures, graphs and tables well characterized clearly showing the proposed data! The results of this important study indicate the need to enforce regulations towards restricting aquarists’ access to threatened and potentially invasive species, as well as measures aimed at informing and raising aquarists’ awareness of conservation measures related to the hobby in Brazil.
The manuscript is very good but needs adjustments throughout the manuscript. The suggestions were indicated with the aim of improving the consistency. All my suggestions are inserted into dialog boxes throughout the text of the manuscript. Authors must verify all sections of the manuscript because I have inserted observations everywhere.

Annotated reviews are not available for download in order to protect the identity of reviewers who chose to remain anonymous.

---

## Round 0.2 · accepted · Accept

Thank you for your hard work on the revisions. It is a pleasure to accept this work and move it into production.

·

Basic reporting

I am satisfied with the explanations and the modifications of the MS.

Experimental design

Clear

Validity of the findings

Valid

Additional comments

I would like to congratulate the authors for a nice paper.

Reviewer 2 ·

Basic reporting

Clear and unambiguous, professional English used throughout.

Experimental design

Research question well defined, relevant & meaningful. It is stated how research fills an identified knowledge gap.

Validity of the findings

All underlying data have been provided; they are robust, statistically sound, & controlled.

Additional comments

I checked the manuscript and found that the authors strictly followed my recommendations. Thus, the manuscript has improved considerably (along with the suggestions of the other reviewer) and I am happy with the final result. Accordingly, I approve the manuscript for publication.
I enjoyed reading this.